# IMAGE GENERATION WITH CHANNEL-WISE QUANTI-ZATION

## ABSTRACT

We present a novel image generation model with channel-wise quantization. Our method quantizes image feature along channel into discrete codes. Then based on the learned codes, our approach adopts masked-prediction paradigm for image generation. Compared with widely used spatial tokenizers, our channel-wise tokenizer has an efficient modeling for image structure and strong representational capacity. Besides, the codebook usage of our tokenizer can reach 100% under different codebook size. Using the channel-wise tokenizer, our generation framework achieves competitive performances on various benchmarks of image generation. In particular, on ImageNet 256x256 benchmark, our method significantly improve baseline by improving Frechet inception distance (FID) to 1.87. Furthermore, we also validate the effectiveness of our proposed method on text-to-image generation.

## 1 INTRODUCTION

Image synthesis has achieved great improvements on quality, diversity and resolution in the past few years. Many prominent frameworks are introduced, such as GAN (Kang et al., 2023), diffusion models (Ho et al., 2020; Rombach et al., 2021; Esser et al., 2024; Li et al., 2024) and VQ models (Van Den Oord et al., 2017; Esser et al., 2020; Yu et al., 2022; Chang et al., 2023). Among these frameworks, VQ models attract enormous attentions, as it is compatible with large language models (LLMs). The training paradigm of VQ models is divided into two stages: learns a compressed discrete representation by a visual tokenizer at the first stage and subsequently learns a underlying data distribution in discrete latent space via a LLMs transformer at second stage. Recent studies (Zheng et al., 2022; Yu et al., 2023; Tian et al., 2024) find that a good compressed discrete representation can improve the upper-bound of image generation.

To learn a good visual representation, existing works (Razavi et al., 2019; You et al., 2022; Huang et al., 2023; Chang et al., 2023; Tian et al., 2024) propose hierarchical tokenizers, which explicitly embed semantic contents and local details, separately. Also, some approaches adopt new objective losses to boost reconstruction quality (Esser et al., 2020) and generation capability of discrete tokens (Gu et al., 2024). These methods learn the compressed visual tokens by **spatially partitioning** the image features, thus the visual tokens focus on local features, leading to strong similarities

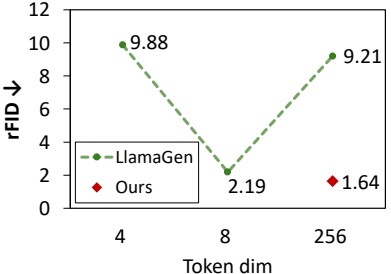

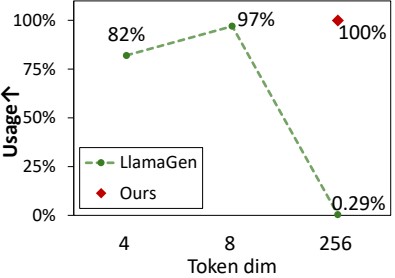

Figure 1: **The comparisons between spatial tokenizer (LlamaGen) and our channel-wise tokenizer** with different token dim.

between tokens and low utilization rate of entire codebook. To achieve a high utilization of codebook, previous studies (Yu et al., 2021; 2023; Sun et al., 2024) decrease the code embedding

dimension. However, decreasing the code embedding reduces the expressive ability of image tokens, thereby the whole capability of an entire codebook also deteriorates.

In this paper, we propose a novel visual tokenizer for image generation, which reaches 100% codebook usage without sacrificing the expressive capacity of tokens, see Figure 1. Specifically, we quantize each channel of image features into a discrete token from the codebook via similarities. Unlike spatially partitioning tokens, the **channel-wise partitioning** tokens possess global structure of the input image and have low similarities between them. Besides, these tokens can capture local details to reconstruct input image. In image generation stage, we adopt the masked language model (MLMs) (Chang et al., 2022; Yu et al., 2023) as the default generator. Utilizing the proposed visual tokenizer and the MLMs, our method can generate high-quality images with small number of sampling steps.

To validate the effectiveness of our method, we conduct extensive experiments on different scenarios. For class-conditional image generation, our approach demonstrates a comparable or superior performance on ImageNet benchmarks. In particular, on ImageNet 256x256 benchmark, our method significantly improve baseline by improving Frechet inception distance (FID) to 2.21 with 634M. To substantiate the transferability of the learned codebook, we also utilize the codebook learned from ImageNet to perform text-to-image generation on COCO dataset. In addition, we conduct ablation studies to show the mechanism behind our proposed tokenizer.

In summary, our contributions are two folds: First, we propose a novel visual tokenizer that channel-wise quantizes image features. Our tokenizer is simple but effective on image tokenization. Besides, due to its 100% codebook usage, our tokenizer is a potential quantizer for training with a large codebook. Second, based on this tokenizer, our generation framework can achieve comparable performance with the state-of-the-art methods on various image generation tasks.

## 2 RELATED WORK

**Image tokenization.** As shown in VQ-VAE (Van Den Oord et al., 2017), image tokenization quantizes image features into discrete tokens derived from a codebook via similarities. To improve image fidelity, VQ-GAN (Esser et al., 2020) applies adversarial loss and perceptual loss in image reconstruction stage. Besides, RQ-VAE (Lee et al., 2022) and MoVQ (Zheng et al., 2022) converts a single index token into a stacked of tokens to reconstruct a high-quality image. Subsequent approaches adopt multi-scale paradigm (Razavi et al., 2019; You et al., 2022; Huang et al., 2023; Chang et al., 2023; Tian et al., 2024) to advance reconstruction quality. VAR (Tian et al., 2024) encodes an image into multi-scale token maps, capturing the global structure and local details. Although these methods have obtained a high image quality, they suffer from low codebook usage with increasing codebook size. To achieve a high utilization of codebook, previous studies (Yu et al., 2021; 2023; Sun et al., 2024) decrease the code embedding dimension, degrading the expressive capacity. Unlike existing works that spatially partitioning images into tokens, our method obtains discrete tokens from image features via channel-wise partitioning. Our method can reach a 100% utilization for the codebook without sacrificing the expressive capacity of tokens.

**Autoregressive models.** With the good discrete tokens, autoregressive models (ARs) (Esser et al., 2020; Lee et al., 2022; Yu et al., 2022; Tian et al., 2024) learn to predict image tokens in an autoregressive manner using a decoder-only transformer. VQ-GAN (Esser et al., 2020) is the first work to employ a decoder-only transformer to generate image tokens for many vector-quantized image modeling tasks. Parti (Yu et al., 2022) is able to generate photorealistic and content-rich images by scaling the transformer up to 20B parameters. VAR (Tian et al., 2024) redefines the autoregressive learning on images as coarse-to-fine "next-scale prediction".

**Masked-predition models.** Unlike autoregressive models, masked-prediction models (Chang et al., 2022; Yu et al., 2023; Chang et al., 2023; Yu et al., 2024) begins with generating all tokens of an image simultaneously and then refines the image iteratively conditioned on the previous generation using a bidirectional transformer decoder. Based on masked-prediction mechanism, MaskGIT (Chang et al., 2022) accelerates autogressive decoding by up to 64x. Due to its efficiency, our method adopt masked-prediction models for image generation stage.

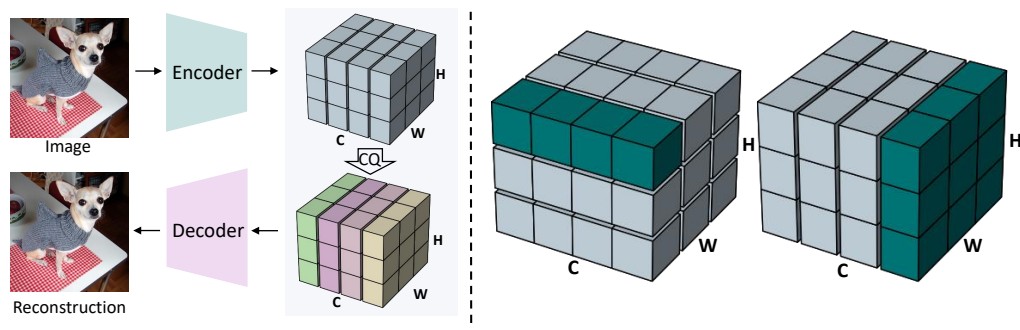

**Channel-wise Quantization**    **Spatial Tokenizer vs. Channel-wise Tokenizer**

Figure 2: **Overview of image quantization in our approach.** The cubes represent the feature tensors, with C as the channel axis, (H, W) as the spatial axes. Left: a quantized autoencoder with our channel-wise quantization. CQ denotes channel-wise quantization. Right: the difference of quantized unit between spatial tokenizer and channel-wise tokenizer. The highlighted pixels (mineral green) are quantized by tokenizers.

## 3  METHOD

### 3.1  PRELIMINARY: SPATIAL TOKENIZER

**Image quantization.**    In VQ models, the goal of image quantization is to learn discrete token representations for image generation stage. Given an image $\mathbf{I} \in \mathbb{R}^{H \times W \times 3}$, encoder $\mathcal{E}$ extracts image features $\mathbf{Z} \in \mathbb{R}^{H_1 \times W_1 \times C}$ with downsample factor $f = H/H_1 = W/W_1$. Then quantizer $\mathcal{Q}$ converts $\mathbf{Z}$ into discrete tokens across **spatial dimension**. For each vector $\boldsymbol{z}_{(i,j)} \in \mathbb{R}^C$ in $\mathbf{Z}$, the quantizer find the closest token index from a codebook $\mathcal{C} \in \mathbb{R}^{K \times C}$ via similarities (e.g. euclidean distance),

$$\mathcal{Q}(\boldsymbol{z}_{(i,j)}; \mathcal{C}) = \underset{k \in [K]}{\arg\min} ||\boldsymbol{z}_{(i,j)} - \boldsymbol{e}_k||_2^2 \tag{1}$$

where $\boldsymbol{e}_k \in \mathcal{C}$. The quantized vector is $\boldsymbol{z}_{(i,j)}^q = \boldsymbol{e}(\mathcal{Q}(\boldsymbol{z}_{(i,j)}; \mathcal{C}))$. $\mathbf{Z}^q \in \mathbb{R}^{H_1 \times W_1 \times C}$ are the quantized features. The decoder $\mathcal{D}$ takes the quantized features as input and output the reconstruction images.

**Discussion.**    Current state-of-the-art tokenizers follow two design rules: (1) high codebook usage; (2) multi-scale feature quantization. For a high codebook utilization, most works reduce the code embedding dimension. This degrades the capacity of image tokens and leads to a poor representation of an whole codebook. To improve the image quality, multi-scale feature quantization is introduced, which captures global structure and local details using different groups of tokens. This hierarchical paradigm leads to a heavy computation cost due to excessive tokens.

### 3.2  CHANNEL-WISE TOKENIZER

Unlike existing spatial tokenizers, we propose channel-wise tokenizer, which quantizes the image features across **channel dimension**, as shown in Figure 2. Concretely, the codebook is $\mathcal{C}' \in \mathbb{R}^{K \times H_1 W_1}$, where the dimension of each code vector is $H_1 W_1$. Given image feature $\mathbf{Z} \in \mathbb{R}^{H_1 \times W_1 \times C}$, it is firstly flattened into a 1D sequence $\mathbf{Z}_c \in \mathbb{R}^{C \times H_1 W_1}$. For each vector $\boldsymbol{z}_c \in \mathbb{R}^{H_1 W_1}$ in $\mathbf{Z}$, the channel-wise quantizer $\mathcal{Q}'$ find the closet token index from a codebook $\mathcal{C}'$ via similarities as follows:

$$\mathcal{Q}'(\boldsymbol{z}_c; \mathcal{C}') = \underset{k \in [K]}{\arg\min} ||\boldsymbol{z}_c - \boldsymbol{e}_k'||_2^2 \tag{2}$$

where $\boldsymbol{e}_k' \in \mathcal{C}'$. The quantized vector is $\boldsymbol{z}_c^q = \boldsymbol{e}(\mathcal{Q}'(\boldsymbol{z}_c; \mathcal{C}'))$. The quantized feature $\mathbf{Z}_c^q \in \mathbb{R}^{C \times H_1 W_1}$ can be converted into $\mathbf{Z}^q \in \mathbb{R}^{H_1 \times W_1 \times C}$ and then mapped into the reconstruction image $\mathbf{I}^q$ by the decoder.

**Training losses.** Following VQ-VAE (Van Den Oord et al., 2017), we use the straight-through estimator (Bengio et al., 2013) to approximate the gradient of the channel-wise quantizer. We apply the reconstruction loss to optimize the encoder and decoder, $\mathcal{L}_{mse} = ||\mathbf{I} - \mathbf{I}^q||_2^2$. For a higher reconstruction quality, we employ perceptual loss (Zhang et al., 2018) and adversarial loss (Goodfellow et al., 2014) with StyleGAN discriminator (Karras et al., 2020). We also use LeCam regularization (Tseng et al., 2021) to stabilize GAN training. For codebook learning, we use the following loss:

$$\mathcal{L}_{codebook} = ||sg[z_c^q] - e_k'||_2^2 + \beta ||z_c^q - sg[e_k']||_2^2 \tag{3}$$

where sg denotes the stopgradient operator. In equation 3, the first term is used to update the codebook and the second term is commitment loss to force the encoder features to be close to codebook embeddings, where $\beta$ is commitment loss weight. We find entropy regularization (Yu et al., 2023; Gu et al., 2024) is bad for our codebook learning, and do not use it.

**Discussion.** For channel-wise tokenizer, its quantized tokens can capture image structures naturally due to its global receptive field. Besides, these tokens need to possess local details for a high-quality image reconstruction. Thus, channel-wise quantized tokens contain global structures and local details at the same time. It is a potential alternative to multi-scale feature quantization used in recent studies (Chang et al., 2023; Tian et al., 2024).

For default spatial tokenizers, the quantized tokens are more like **visual characters**, since they pay attention to local image areas and are easily to collapse into a limited number of visual units. Conversely, the quantized tokens in channel-wise tokenizer represent an image from an overall perspective. The image tokens concentrate on larger image areas and more diverse than ones generated by spatial tokenziers, thus we denote them as **visual words**. As a result, the codebook usage for our channel-wise tokenizer reaches 100%.

### 3.3 MASKED CHANNEL-WISE PREDICTION

Inspired by MaskGiT (Chang et al., 2022), we learn the distribution priors of the channel-wise visual tokens using a bidirectional transformer for image generation. Specifically, in each training step, we sample a subset of tokens and replace them with a special mask token. Then, based on the masked token sequence, we employ a directional transformer to predict the corresponding discrete token index of those masked tokens. In the inference, we generate all tokens in the image simultaneously in a single pass and then select the predictions of the masked tokens with high confidence to update the masked images. Through this way, we can refine the image tokens iteratively conditioned on the previous generation and output the full generated tokens, which are later mapped to image pixels.

In addition, to improve the training stability, we adopt query-key normalization with the RMSNorm (Zhang & Sennrich, 2019). As done in MaskGiT (Chang et al., 2022), we train our model with a variable masking rate based on a Cosine scheduling for a high quality of image generation.

**Classifier-free guidance.** Classifier-free guidance (Ho & Salimans, 2022) is an useful technique to improve generation quality and text-image alignment, thus we employ it in our masked channel-wise prediction. At training time, we drop text conditioning on 10% of samples randomly and replace it with a null embedding. In the inference, we compute a conditional logit $\ell_c$ and an unconditional logit $\ell_u$ for each masked token. We compute the final logit $\ell_g$ as follows:

$$\ell_g = t\ell_c - (t-1)\ell_u \tag{4}$$

where $t$ is the guidance scale.

## 4 EXPERIMENTS

### 4.1 EXPERIMENTAL SETUPS

**Datasets.** For image tokenizer and class-conditional image generation, we use ImageNet (Deng et al., 2009) at $256 \times 256$ and $512 \times 512$ resolutions, which contains 1,281,167 training images and 50,000 validation images from 1K different classes. Following U-ViT (Bao et al., 2022), for text-to-image generation, we train the generator only using MS-COCO at $256 \times 256$ resolution,

Table 1: **Model sizes and architecture configurations of our models.** The configurations are following previous works (Chang et al., 2022).

| Model | #Para. | #heads | #layers | Hidden size | MLP dim |
|---|---|---|---|---|---|
| Ours-B | 305M | 16 | 24 | 1024 | 4096 |
| Ours-L | 634M | 16 | 32 | 1280 | 5120 |
| Ours-H | 1.0B | 16 | 36 | 1536 | 6144 |

Table 2: **Class-conditional image generation on ImageNet 256×256.** "Tokenizer Type": the type of quantizer used by generative models, "S" denotes "spatial quantizer", "C" denotes "channel-wise quantizer". "↓" or "↑" indicate lower or higher values are better. Metrics include Fréchet inception distance (FID), inception score (IS). "#Para": the model size used in image generation. "#Step": the number of model runs needed to generate an image. $^\dagger$: codebook size is 65536.

| Type | Models | Tokenizer Type | FID↓ | IS↑ | #Para | #Step |
|---|---|---|---|---|---|---|
| GAN | BigGAN (Brock, 2018) | - | 6.95 | 224.5 | 112M | 1 |
| | GigaGAN (Kang et al., 2023) | - | 3.45 | 225.5 | 569M | 1 |
| | StyleGan-XL (Sauer et al., 2022) | - | 2.30 | 265.1 | 166M | 1 |
| Diffusion | ADM (Dhariwal & Nichol, 2021) | - | 10.94 | 101.0 | 554M | 250 |
| | CDM (Ho et al., 2022) | - | 4.88 | 158.7 | - | 8100 |
| | LDM-4-G (Rombach et al., 2021) | - | 3.60 | 247.7 | 400M | 250 |
| | DiT-L/2 (Peebles & Xie, 2022) | - | 5.02 | 167.2 | 458M | 250 |
| | DiT-XL/2 (Peebles & Xie, 2022) | - | 2.27 | 278.2 | 675M | 250 |
| AR | VQGAN (Esser et al., 2020) | S | 15.78 | 74.3 | 1.4B | 256 |
| | ViTVQ (Yu et al., 2021) | S | 4.17 | 175.1 | 1.7B | 1024 |
| | RQTran. (Lee et al., 2022) | S | 7.55 | 134.0 | 3.8B | 68 |
| | LlamaGen-L (Sun et al., 2024) | S | 3.80 | 248.28 | 343M | 256 |
| | LlamaGen-XL (Sun et al., 2024) | S | 3.39 | 227.08 | 775M | 256 |
| | LlamaGen-XXL (Sun et al., 2024) | S | 3.09 | 253.61 | 1.4B | 256 |
| | Open-MAGVIT2-B (Luo et al., 2024) | S | 3.08 | 258.26 | 343M | 256 |
| | Open-MAGVIT2-L (Luo et al., 2024) | S | 2.51 | 271.70 | 804M | 256 |
| | Open-MAGVIT2-XL (Luo et al., 2024) | S | 2.33 | 271.77 | 1.5B | 256 |
| VAR | VAR-d16 (Tian et al., 2024) | S | 3.60 | 257.5 | 310M | 10 |
| | VAR-d20 (Tian et al., 2024) | S | 2.95 | 306.1 | 600M | 10 |
| | VAR-d24 (Tian et al., 2024) | S | 2.33 | 320.1 | 1.0B | 10 |
| | VAR-d30 (Tian et al., 2024) | S | 1.97 | 334.7 | 2.0B | 10 |
| Mask. | MaskGIT (Chang et al., 2022) | S | 6.18 | 182.1 | 227M | 8 |
| | RCG (cond.) (Li et al., 2023) | S | 3.49 | 215.5 | 502M | 20 |
| | MagViT-2 (Yu et al., 2023) | S | **1.78** | 319.4 | 307M | 64 |
| Mask. | Ours-B | C | 2.77 | 305.3 | 305M | 10 |
| | Ours-L | C | 2.46 | 302.5 | 634M | 10 |
| | Ours-H | C | 2.39 | 338.2 | 1.0B | 10 |
| | Ours-B* | C | 2.21 | 301.2 | 305M | 64 |
| | Ours-L* | C | 2.02 | 323.4 | 634M | 64 |
| | Ours-H* | C | 1.91 | **344.9** | 1.0B | 64 |
| | Ours-L$^\dagger$ | C | 1.87 | 320.4 | 634M | 64 |

which contains 82,783 training images and 40,504 validation images. Each image is annotated with 5 captions.

**Architecture configurations.** For channel-wise tokenizer, we follow the implementation of VG-GAN (Esser et al., 2020). For simplicity, we remove the attention blocks from the architecture of

Table 3: **Class-conditional image generation on ImageNet 512×512.** "Tokenizer Type": the type of quantizer used by generative models, "S" denotes "spatial quantizer", "C" denotes "channel-wise quantizer".

| Type | Models | Tokenizer Type | FID↓ | IS↑ | #Para | #Step |
|------|--------|----------------|------|-----|-------|-------|
| GAN | BigGAN (Brock, 2018) | - | 8.43 | 177.9 | - | 1 |
| Diffusion | ADM (Dhariwal & Nichol, 2021) | - | 23.24 | 101.0 | 559M | 250 |
| | DiT-XL/2 (Peebles & Xie, 2022) | - | 3.04 | 240.8 | 675M | 250 |
| AR | VQGAN (Esser et al., 2020) | S | 26.52 | 66.8 | 227M | 1024 |
| VAR | VAR-d36-s (Tian et al., 2024) | S | 2.63 | 303.2 | >2B | 10 |
| Mask. | MaskGiT (Chang et al., 2022) | S | 7.32 | 156.0 | 227M | 12 |
| | MagViT-v2 (Yu et al., 2023) | S | **1.91** | 324.3 | 307M | 64 |
| Mask. | Ours-B | C | 2.68 | 318.5 | 305M | 10 |
| | Ours-L | C | 2.46 | 336.4 | 634M | 10 |
| | Ours-B* | C | 2.22 | 323.4 | 305M | 64 |
| | Ours-L* | C | 2.01 | **341.5** | 634M | 64 |

channel-wise tokenizer. As suggested in VIT-VQGAN (Yu et al., 2021), we employ the StyleGAN discriminator in the training. Note that for stable training, we disable the default fp16 training for StyleGAN discriminator.

Following MaskGiT (Chang et al., 2022), we adopt a bidirectional transformer for masked visual modeling. As shown in Table 1, the base and large model have 305M and 634M parameters, respectively. In text-to-image generation, we convert discrete texts to a sequence of embeddings using a CLIP text encoder following Stable Diffusion (Rombach et al., 2021).

**Training.** Following SeQ-GAN (Gu et al., 2024), we train the channel-wise tokenizer using Adam optimizer (Kingma, 2014). Besides, we train the model for 300 epochs with a total 256 batch size. The initial learning rate is $1e\text{-}4$ and decays to $5e\text{-}5$ via cosine decay schedule. For StyleGAN discriminator, we enable it after training 10 epochs.

The training settings of masked visual transformer in class-conditional image synthesis is as follows: a initial $4e\text{-}4$ learning rate, AdamW optimizer with $\beta_1 = 0.9$, $\beta_2 = 0.96$, and a total 1024 batch size for 1200 epochs. But, in text-to-image tasks, we modify some settings: a initial $1e\text{-}4$ learning rate and a total 256 batch size for 3000 epochs.

**Evaluation metrics.** For image reconstruction, we adopt the reconstruction-FID (rFID), codebook usage, PSNR and SSIM to measure the quality of reconstructed images on ImageNet 50K validation set. To assess class-conditional image generation, we calculate Fréchet inception distance (FID) (Heusel et al., 2017) and Inception score (IS) (Salimans et al., 2016) on ImageNet using 50K generated images compared against the ImageNet training set. For text-to-image evaluation, we randomly draw 30K prompts from the MS-COCO validation set, and generate samples on these prompts to compute FID.

### 4.2 CLASS-CONDITIONAL GENERATION

**Setup.** We test our models with two variants (305M and 634M) on ImageNet class-conditional generation benchmarks and compare them with the state-of-the-art image generation model families. Unlike existing VQVAE-based models, our models are based on channel-wise tokenizer. Note that we train the tokenzier directly on ImageNet, while VAR (Tian et al., 2024) and VQGAN (Esser et al., 2020) use OpenImages (Kuznetsova et al., 2020) as training data for VQVAE. The results are demonstrated in Table 2 and Table 3.

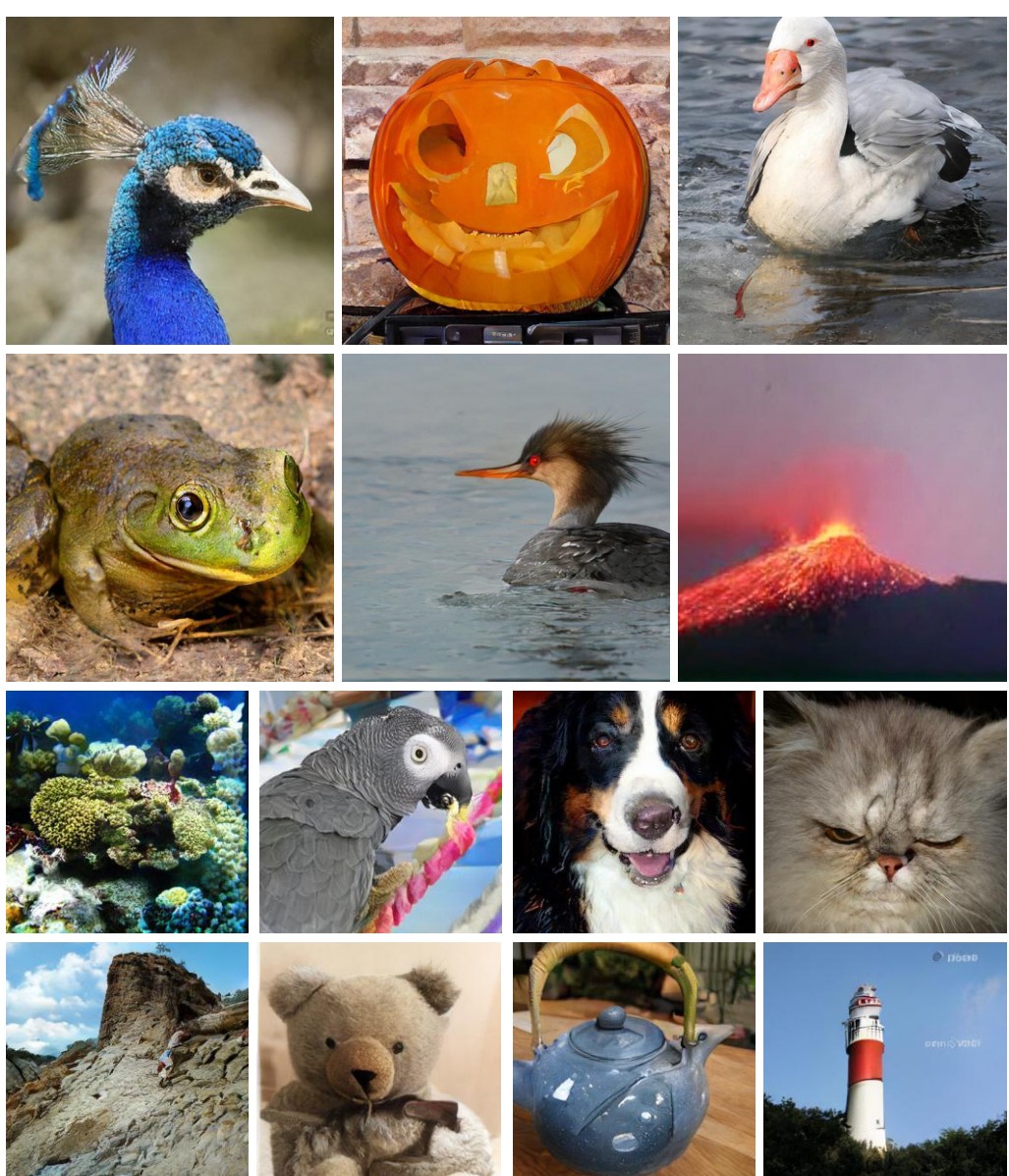

Figure 3: **Generated samples from our proposed models trained on ImageNet.** We show 512×512 samples (top-2 rows) and 256×256 samples (bottom-2 rows). The samples are generated with Our-L models with 10 steps.

**Results.** In comparison with existing generative methods, our method establishes a new model class based on channel-wise tokenizer. As shown in Table 2, under the same settings, our approach achieves better FID and IS than generative adversarial networks (GAN), diffusion models (Diffusion), autoregressive model (AR), visual autoregressive (VAR) and masked-prediction models (Mask.), except for MagViT-2 (Yu et al., 2023). In particular, our method achieves a highest IS score among all methods. Note that MagViT-2 utilizes a larger codebook than ours, thus it is reasonable for them to achieve better FID score than ours. We must point out that our method have potential to obtain better performance with a large-scale codebook, see section 4.4 for more details. But due to limited compute resources, we leave it for the future. In addition, the effectiveness of out model is also validated on the 512×512 synthesis benchmark, as shown in Table 3. Our model out-

Table 4: **Text to image generation on MS-COCO 256×256 validation.** The evaluations are on COCO 30k val2014 set at 256×256 resolution.

| Models | Type | Training datasets | FID↓ |
|---|---|---|---|
| LAFITE (Zhou et al.) | GAN | CC3M (3M) | 26.94 |
| Make-A-Scene (Gafni et al., 2022) | AR | Union datasets (35M) | 11.84 |
| DALL-E 2 (Ramesh et al., 2022) | Diffusion | DALL-E dataset (250M) | 10.39 |
| Imagen (Saharia et al., 2022) | Diffusion | Internal dataset (460M) + LAION (400M) | 7.27 |
| Re-Imagen (Chen et al., 2022) | Diffusion | KNN-ImageText (50M) | 6.88 |
| XMC-GAN (Zhang et al., 2021) | GAN | MS-COCO (83K) | 9.33 |
| Friro (Fan et al., 2023) | Diffusion | MS-COCO (83K) | 8.97 |
| U-ViT-S/2 (Bao et al., 2023) | Diffusion | MS-COCO (83K) | 5.95 |
| Ours-L | Mask. | MS-COCO (83K) | 6.40 |
| Ours-L* | Mask. | MS-COCO (83K) | **5.85** |

Table 5: **Comparisons with other image tokenizers.** The evaluations are on ImageNet 50k validation set and COCO 5k val2017 set at 256×256 resolution. The compression ratio is 16.

| Method | #Tokens | #dim | size | ImageNet | | | MS-COCO | | |
|---|---|---|---|---|---|---|---|---|---|
| | | | | rFID↓ | PSNR↑ | SSIM↑ | rFID↓ | PSNR↑ | SSIM↑ |
| VQGAN | 256 | 256 | 1024 | 8.30 | 19.51 | 0.614 | 16.95 | 19.08 | 0.613 |
| VQGAN | 256 | 256 | 16384 | 4.99 | 20.00 | 0.629 | 12.29 | 19.57 | 0.630 |
| MaskGIT | 256 | 256 | 1024 | 2.28 | - | - | - | - | - |
| LlamaGen | 256 | 256 | 16384 | 9.21 | 18.32 | 0.575 | - | - | - |
| LlamaGen | 256 | 8 | 16384 | 2.19 | 20.79 | 0.675 | 8.11 | 20.42 | 0.678 |
| Ours | 256 | 256 | 16384 | 1.64 | 18.72 | 0.866 | 7.95 | 17.92 | 0.860 |
| Ours | 512 | 256 | 16384 | **0.98** | **21.47** | **0.925** | **6.22** | **21.10** | **0.931** |

performs other methods by a large margin on both FID and IS, except for MagViT-2. In particular, ours-L* performs better on FID than VAR with > 2B parameters and beats MagViT-2 on IS score. In Figure 3, we show the generated samples on ImageNet at 512×512 and 256×256 resolutions.

### 4.3 TEXT-TO-IMAGE GENERATION

**Setup.** We evaluate our model for text-to-image generation on the standard benchmark dataset MS-COCO. We train masked-prediction model with MS-COCO 256×256 training data following U-ViT (Bao et al., 2023). Note that we use the tokenzier trained on ImageNet for image quantization, which does not utilize large-scale external dataset to train.

**Results.** As shown in Table 4, Our-L outperforms most of existing methods, such as Re-Imagen and Friro. By further increasing the sampling steps from 10 to 64, Our-L* can even obtained 5.85 FID on MS-COCO benchmark, achieving a better result than U-ViT-S/2. These results demonstrate the effectiveness of our method on text-to-image generation.

### 4.4 ABLATION STUDY

**Comparisons with other image tokenizers.** We compare with other image tokenizers, including VQGAN (Esser et al., 2020), MaskGIT (Chang et al., 2022) and LlamaGen (Sun et al., 2024). As shown in Table 5, our tokenizer have the best rFID score among these tokenizers. Since it is a channel-wise quantizer and can capture global structure, our tokeniezr demonstrates superior results on SSIM score. We find our tokenizer lag behinds on PSNR score due to limited tokens. When increasing the number of tokens, our tokenizer achieves a large gain on PSNR. Besides, it also obtains higher performances on rFID and SSIM.

Table 6: **Ablation studies on tokenizers design.** The evaluations are on ImageNet 50k validation set at 256×256 resolution. The default number of image tokens is 256. The compression ratio is 16.

(a) Spatial quantizer vs. Channel-wise quantizer.

| Method | Token dim | rFID↓ | Usage↑ |
|--------|-----------|-------|--------|
| LlamaGen | 256 | 9.21 | 0.29% |
|  | 32 | 3.22 | 20.9% |
|  | 8 | 2.19 | 97.0% |
|  | 4 | 9.88 | 82.0% |
| Ours | 256 | **1.64** | 100% |

(b) Effect of codebook size.

| Codebook size | rFID↓ | Unique ratio | Usage↑ |
|---------------|-------|--------------|--------|
| 1024 | 2.25 | 72.5% | 100% |
| 16384 | 1.64 | 92.6% | 100% |
| 65536 | 1.43 | 95.6% | 100% |
| 131072 | 1.33 | 96.9% | 100% |

**Generalization of our tokenizer.** To validate the generalization of our tokenizer, we directly evaluate our tokenizer trained with ImageNet on MS-COCO of 256×256 image resolution. Note that MS-COCO mainly have scene-centric images, while ImageNet focuses on object-centric images. There are a big domain gap between these two datasets. As shown in Table 5, compared with other tokenizers, our tokenizer achieves the best rFID score and SSIM score. The results showcase that our tokenizer is a generalizable image tokenizer.

**Effect of tokenizer design.** We compare our tokenizer with the spatial tokenzier used in LlamaGen. For fair comparison, we use the same codebook size (16384) for these two tokenizers. As shown in Table 6a, LlamaGen can reduce the token dim to improve the reconstruction quality and codebook usage. However, reducing token dimension degrades the expressive capacity of the codebook. Unlike spatial tokenzier, our tokenizer can achieve a better image quality and codebook usage without sacrificing the expressive capacities of quantized tokens. This demonstrates that our tokenizer is a potential tokenizer for image quantization.

In addition, we compare the performance of our tokenizer with different codebook sizes. To understand our tokenizer deeply, we also propose a new metric: unique ratio. For an image, we calculate the ratio of unique tokens in the total image tokens. The high ratio of unique tokens means the more distinct image features that tokenizer captures. As shown in Table 6b, with increasing codebook size, the rFID score is getting better for our tokenzier. Meanwhile, the codebook usage of our tokenzier reaches 100% under all codebook sizes. This demonstrates that the effectiveness of our tokenizer with increasing codebook size. We also find that the unique ratio increases with larger codebook. With a small codebook, our tokenizer have no enough capacity to represent image in details, while our tokenizer can capture more image features when using a large codebook.

**Ablation studies on image generation.** To study our model on image generation, we analyze the effects of different components, including model size, codebook size and sampling steps. As demonstrated in Table 7, we find that the model with 64 sampling steps boost generation performance largely. The codebook size also has a positive benefit to performance. With larger codebook, our model boosts FID to 1.87. This demonstrates our method can obtain better performance with increasing codebook size.

Table 7: **Ablation studies on image generation.**

| Model size | Codebook size | #Step | FID↓ | IS↑ |
|------------|---------------|-------|------|------|
| 305M | 16384 | 10 | 2.77 | 305.3 |
|  |  | 64 | 2.21 | 301.2 |
|  | 65536 | 10 | 2.53 | 301.1 |
|  |  | 64 | 2.04 | 306.9 |
| 634M | 16384 | 10 | 2.46 | 302.4 |
|  |  | 64 | 2.01 | **323.4** |
|  | 65536 | 10 | 2.34 | 312.2 |
|  |  | 64 | **1.87** | 320.4 |

## 5 LIMITATIONS

There are two limitations in our method. First, our channel-wise tokenizer needs to be trained separately for different image resolution. Conversely, spatial tokenizers are trained in a low resolution once but directly used for various high image resolution, though the transfer performance is not

optimal. Second, due to limited compute resources, we can not train a large model to validate the scalability of our approach.

## 6 CONCLUSION

We introduce a novel image generation model with channel-wise tokenizer. The channel-wise tokenizer provides a novel image quantization and achieves superior performance on image reconstruction. Compared with widely used spatial tokenizer, it showcases a high codebook usage. With the proposed channel-wise tokenizer, our generation framework can perform comparable performance with state-of-the-art models on image generation, demonstrating the effectiveness of our proposed method.

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

# A APPENDIX

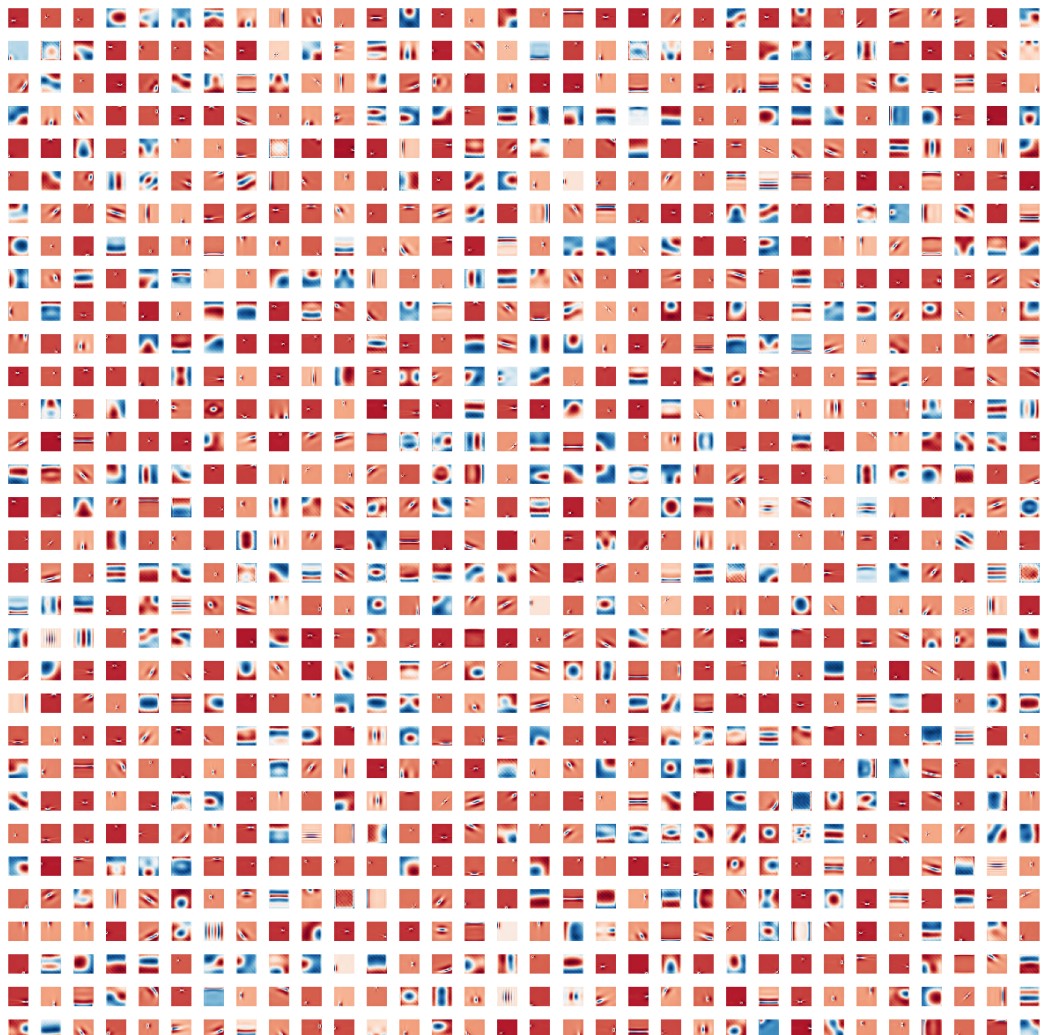

Figure 4: **Visualization of codebook tokens (size=1024) learned by our channel-wise tokenizer.** The red and blue means high and low activations, respectively.

## A.1 CODEBOOK ANALYSIS

We visualize codebook tokens (size=1024) in Figure 4. The visualization shows that our channel-wise tokenizers can capture image structure and local details simultaneously. For example, in row 1, the first three tokens show high activations on global areas. It means that these three tokens are used to represent image structure. The 4th, 5th, and 6th tokens have high activations on local areas, meaning they are used for local details.

