# OpenReview forum: "Image Generation with Channel-wise Quantization"
_ICLR.cc/2025/Conference — Submitted to ICLR 2025_

### Official Review · Reviewer_6xWH · 2024-10-25

**Soundness:** 2
**Presentation:** 2
**Contribution:** 2
**Rating:** 3
**Confidence:** 4

**Summary:**

This work presents a novel image generation model that utilizes channel-wise quantization to convert image features into discrete codes along the channel dimension, adopting a masked prediction paradigm for image generation. This approach offers efficient modeling of image structures and strong representational capacity, outperforming or matching state-of-the-art methods on the ImageNet benchmark. The authors also performed the text-to-image generation and demonstrated transferability to text-to-image generation on the COCO dataset. The contributions of this work include a simple yet effective visual tokenizer with 100% codebook usage and a generation framework based on channel-wise quantization for image generation tasks.

**Strengths:**

This paper is easy to follow and includes comprehensive experiments to demonstrate the effectiveness of the proposed method. It proposes novel channel-wise quantization, which offers efficient modeling of image structures and strong representational capacity. It also proposed a simple yet effective tokenizer with 100% codebook usage. These features enable the whole image generation framework to achieve superior or comparable performance to state-of-the-art methods across various image generation tasks.

**Weaknesses:**

1) This paper's biggest contribution is proposing a channel-wise quantization for image generation. However, channel-wise quantization is widely used in various applications, such as classification [a], LLM compression [b], and super-resolution [c].

[a] RDO-Q: Extremely Fine-Grained Channel-Wise Quantization via Rate-Distortion Optimization. ECCV 2022.

[b] OutlierTune: Efficient Channel-Wise Quantization for Large Language Models. Arxiv 2024.

[c] DAQ: Channel-Wise Distribution-Aware Quantization for Deep Image Super-Resolution Networks. WACV 2022.

2) Why not combine SD series for comparison? For example, SD1.5, SDXL and SD2.1.

3) More experimental results about high-resolution image generation should be provided to demonstrate the effectiveness of the proposed method.

4) The authors did not evaluate the text-image alignment between the text prompts and the synthesized images.

5) It is not reasonable to develop a channel-wise tokenizer, which is limited to one specific image resolution. One general channel-wise tokenizer algorithm that could be suitable for all image resolutions is favored. Such limitation heavily limits the contribution of this work.

6) The authors only conducted experiments on MS-COCO and ImageNet datasets with the image resolution 256*256.

7) Marginal performance. Compared with VAR, the proposed method only shows very marginal performance improvement (or even a slight performance drop) under the same experimental setting: similar network parameters and inference step from Table 2.

Minor issues:

1) The best results under the same setting in Table 2 should be bold.

2) Figure 2 is a little blur and seems that some parts are screenshots of other images.

**Questions:**

What is the purpose of the image generation?

Only evaluating the image quality of the synthesized images based on evaluation metrics like FID, SSIM and PSNR is not enough. Can the synthesized images promote the downstream visual perception performance such as classification?

---

> ### Author Response · Authors · 2024-11-14
> **rebuttal**
>
> Thank you for your careful review and helpful suggestions!
>
> **Q1: However, channel-wise quantization is widely used in various applications, such as classification [a], LLM compression [b], and super-resolution [c].**
>
> **A1:** I disagree with this statement. The motivation and methodolgy are different. The channel-wise quantization used in [a], [b] and [c] belong to **model quatization**, which is **a technique to accelerate model inference**. While, the channel-wise quantization for image generation aims to **learn visual representation**.
>
> **Q2: Why not combine SD series for comparison? For example, SD1.5, SDXL and SD2.1.**
>
> **A2:**  As shown in section 4.3, we train our model only with MS-COCO (83K) dataset following U-ViT. However, SD series are trained with large-scale datasets. It is unfair to compare with SD series directly, due to the large data gap. Besides, to our knowledge, the FID of SDv2.1 on COCO is 12.43, refer to [1]. It lags behind our model (5.85).
>
> **Q3&Q6: More experimental results about high-resolution image generation should be provided to demonstrate the effectiveness of the proposed method. The authors only conducted experiments on MS-COCO and ImageNet datasets with the image resolution 256*256.**
>
> **A3&A6:** As shown in Table 3, we provide the class-conditional image generation on ImageNet 512x512, not only the image resolution 256x256. The results on ImageNet 512x512 demonstrate the effectiveness of the proposed method. For higher image generation, like 1024x1024, it spends much time to implement, since we need to train channel-wise tokenzier for 1024x1024. Besides, there is an option to implement image generation on 1024x1024. We can use a super resolution model to upscale image from 512x512 to 1024x1024. We leave it for future work.
>
> **Q4: The authors did not evaluate the text-image alignment between the text prompts and the synthesized images.**
>
> **A4:** We are conducting the evaluation on text-image alignment. We will update results as soon as possible.
>
> **Q5: It is not reasonable to develop a channel-wise tokenizer, which is limited to one specific image resolution. One general channel-wise tokenizer algorithm that could be suitable for all image resolutions is favored. Such limitation heavily limits the contribution of this work.**
>
> **A5:** I disagree with this statement. First, for our method, it is an alternative to scale codebook. For codebook scaling, we conduct an experiment with codebook size 2^17=131072 without entropy regularization. The model obtains 1.33 rFID and 100\% usage on ImageNet. This result validates the advantage of our method on codebook scaling. Second, **although a model is trained for one specific image resolution, we can train different models for any image resolution.**  Our method is not limited to one specific image resolution, as shown in Table 2 and Table 3.
>
> **Q7: Marginal performance.**
>
> **A7:** We update our results in the paper. With widely used sampling scheme, our models achieve higher performance than VAR on FID and IS scores, as shown in Table 1 and Table 2.
>
> **Q8: The best results under the same setting in Table 2 should be bold.**
>
> **A8:** Thanks for your suggestion. We have updated it in our paper.
>
> **Q9: Figure 2 is a little blur and seems that some parts are screenshots of other images.**
>
> **A9:** Thanks for your suggestion. We will update it in the final version.
>
> **Q10: What is the purpose of the image generation? Only evaluating the image quality of the synthesized images based on evaluation metrics like FID, SSIM and PSNR is not enough. Can the synthesized images promote the downstream visual perception performance such as classification?**
>
> **A10:** I disagree with this statement. In image generation field, almost all works evaluate the image quality of the synthesized images based on evaluation metrics like FID, SSIM and PSNR. Thus, it is enough to validate the effectiveness of our method using these metrics.  Using the synthesized images to promote the downstream visual perception performance is an interesting future work.
>
> [1] Dong, Runpei, et al. "Dreamllm: Synergistic multimodal comprehension and creation." arXiv preprint arXiv:2309.11499 (2023).

---

> ### Author Response · Authors · 2024-11-19
> **rebuttal**
>
> **Q4: The authors did not evaluate the text-image alignment between the text prompts and the synthesized images.**
>
> **A4**: we use CLIP score to show text-image alignment performance. The results are as follows:
>
> **MS-COCO 256x256**
> |   Models    | FID | CLIPScore | #step|
> | ----------- | ----------- |----------- | ----------- |
> |U-ViT-S/2 | 5.95 | **0.296** |  - |
> |Ours-L | 6.40 | 0.288 | 10 |
> |Ours-L*  | **5.85** | 0.289 |  64 |
>
> To our knowledge, the CLIP score of our method shows a good text-image alignment performance.

---

> > ### Comment · Reviewer_6xWH · 2024-11-25
> > **Official Comment by Reviewer 6xWH**
> >
> > I appreciate the author's effort to address my questions. However, after reading the rebuttal, I still have concerns about the motivation and overall design of this work.
> >
> > ## Regarding the overall design
> >
> > The authors claimed, "although a model is trained for one specific image resolution, we can train different models for any image resolution." However, training different models for any image resolution is still expensive and time-consuming. I personally suggest two alternative ways for this problem: 1) proposing a general image tokenizer for all image resolutions or 2) proposing some plug-and-play module for various image resolutions in a training-free manner.
> >
> > I also read other reviews, especially comments from reviewer FnTD. I agree with his/her concern. The proposed method has a limited generalizability to other image resolutions or models.
> >
> > ## Regarding the text-image alignment
> >
> > the improvement of CLIP score over the U-ViT-S method is marginal, which cannot say the proposed method has a better text-image alignment over existing methods.
> >
> > ## Regarding the evaluation of synthesized images
> >
> > My original purpose is to encourage the authors to add more experimental results to demonstrate the effectiveness of the image synthesis to better support their work, making the whole work more solid and comprehensive.
> >
> > To sum up, I feel the overall design of the chanel-wise quantization is questionable. Therefore, I tend to keep my rating at 3.

---

> > > ### Author Response · Authors · 2024-11-25
> > >
> > > **Q1: Regarding the overall design**
> > >
> > > **A1:** We admit that the tokenizer trained on one specific resolution has limitation to generalize to other resolution. But we still stress that although a model is trained for one specific image resolution, we can train different models for any image resolution. Our method is not limited to one specific image resolution, as shown in Table 2 and Table 3. **Our method means the channel-wise quantization.**
> > >
> > > **Q2: Regarding the text-image alignment**
> > >
> > > **A3:** First, in our rebuttal, we claim that **the CLIP score of our method shows a good text-image alignment performance.** We do not say “the proposed method has a better text-image alignment over existing methods.”
> > >
> > > Second, the reason that our method does not obtain better text-image alignment is as follows:  the vae used in U-ViT-S is trained with larger datasets, while our tokenizers is only trained with ImageNet. Thus, the learned concepts are limited for our tokenizers.
> > >
> > > **Q3: Regarding the evaluation of synthesized images**
> > >
> > > **A3:** We have provided the results on ImageNet 256x256, Imagenet 512x512 and MS-COCO 256x256. We also conduct ablation studies on different model sizes (300M, 600M, 1B), codebook size (1024, 16384, 65536, 131072) and channel size (256, 512). These results demonstrate the effectiveness of our method.  **But in your opinion, these results are still not enough.** We admit that using the synthesized images to promote the downstream visual perception performance is an interesting future work. However, this is not a main metric in image generation filed now.

---

> > > > ### Comment · Reviewer_6xWH · 2024-11-25
> > > > **Second response from reviewer 6xWH**
> > > >
> > > > Dear authors, thank you for your response. I would like to offer the following suggestions regarding your rebuttal:
> > > >
> > > > 1) **Engage Constructively with the Reviewer's Feedback**. It is important to avoid directly arguing with the reviewer. Instead, try to understand their comments from their perspective and address their concerns thoughtfully. In our previous discussions, I have questioned the contribution of your work in terms of its generalization ability and its impact on the broader field of image generation. The reviewer's primary concern appears to be that your design may not effectively solve a highly useful or valuable problem due to its restricted setting. Simply reiterating your original points without engaging with their critique may not persuade the reviewer to reconsider their evaluation.
> > > >
> > > > 2) **Address the Suggestion**. The reviewer has suggested that the authors could add some downstream vision tasks to strengthen your work. From the standpoint of the reviewer, this is a reasonable request. If you choose not to incorporate this suggestion, it would be beneficial to provide a clear and well-supported explanation as to why this addition may not be appropriate or feasible. Rather than stating, "However, this is not a main metric in the image generation field now," consider substantiating your position with experimental results or empirical analysis. This approach is more persuasive than relying on the argument that "others did not do this, either."

---

> > > > > ### Author Response · Authors · 2024-11-25
> > > > >
> > > > > Thanks for your comments.
> > > > >
> > > > > **A1:** We admit that our method needs to train different models for different images. It is a limit of our method. But we must stress the contribution of our method: 1) it is a alternative to scaling codebook; 2) it capture global and local information simultaneously. 3) the performance on image generation is superior compared with VAR. We hope the reviewers re-evaluate the contribution of our method. We will address this limit and leave it for the future work.
> > > > >
> > > > > **A3:** We thanks for your suggestion. We have mentioned that using the synthesized images to promote the downstream visual perception performance is an interesting future work. With the existing results, we believe that the effectiveness of our method is validated.

---

### Official Review · Reviewer_xTUG · 2024-10-31

**Soundness:** 2
**Presentation:** 3
**Contribution:** 2
**Rating:** 5
**Confidence:** 4

**Summary:**

In this paper, the authors propose a new image tokenization method called channel-wise quantization, which quantizes image feature along channel. Then, based on the learned tokenizer, the authors use masked-prediction paradigm similar to MaskGIT to generate images. Experiments show that the proposed method achieves competitive performances compared to other image tokenizers and generative models. Besides, the proposed method can reaches 100% codebook usage under different codebook size.

**Strengths:**

+ The writing is clear and easy to follow.
+ The idea of channel-wise quantization is novel and interesting.
+ The reconstruction ability (especially rFID and SSIM) is greatly improved compared to spatial tokenizers.

**Weaknesses:**

- The motivation of proposing channel-wise quantization is not very strong to me. While spatial tokenizers often suffer from low codebook usage and reduced code embedding dimension limits expressive ability, it's unclear how these issues lead to the design of channel-wise quantization.

- The paper lacks a detailed analysis of how the channel-wise tokenizer behaves differently from spatial tokenizers. For example, the authors claim that channel-wise tokens capture both global structures and local details, but there are no direct experiments to support this. It would be interesting if the authors could visualize the learned channel-wise tokens and discuss each channel's representation, so that we can have a deeper understanding of the channel-wise tokenizer.

- The authors attribute the 100% codebook usage to the nature of channel-wise quantization. However, I note that entropy regularization, which is known to be helpful for increasing codebook usage, is adopted in codebook learning. Additionally, the compared method LlamaGen did not use entropy regularization. Thus, I'm not sure if channel-wise quantization is the main factor behind high codebook usage.

**Questions:**

+ As mentioned in weaknesses part, will the channel-wise quantization still reach 100% codebook usage without entropy regularization?

+ What does a channel token typically represent? Is it possible to visualize the learned channel tokens?

---

> ### Author Response · Authors · 2024-11-14
> **rebuttal**
>
> Thank you for your careful review and helpful suggestions!
>
> **Q1: The motivation of proposing channel-wise quantization.**
>
> **A1:** As shown in section 3.2 Discussion, spatial tokenizers pay more attention to local image areas and are easily to collapse into a limited number of visual units. If image tokens concentrate on larger image areas, image tokens can be more diverse, resulting in high usage for codebook. For channel-wise tokenizer, it forces image tokens to focus on large image areas, due to its large receptive filed. Thus, channel-wise tokenizer is an alternative to codebook scaling.
>
> **Q2: The paper lacks a detailed analysis of how the channel-wise tokenizer behaves differently from spatial tokenizers.**
>
> **A2:**
>
> **Image structures:** As shown in “Comparisons with other image tokenizers”, our tokenizer demonstrates superior results on SSIM score than spatial tokenizer. The higher SSIM score shows that our channel-wise tokenizer captures image structure.
>
> **local details:** As well known, IS score is used to evaluate the quality of generated image. Higher IS score means the more details of the generated images. As shown in Table 2 and Table 3, our models achieve higher scores on IS than all previous works. These results show that our channel-wise tokenizer captures local details.
>
> **Q3: Entropy regularization**
>
> **A3:** To validate our channel-wise quantization on codebook usage, we conduct a experiment using codebook size 2^17=131072 without entropy regularization. The model obtains 1.33 rFID and 100\% usage on ImageNet. This result validates that channel-wise quantization is the main factor behind high codebook usage.
>
> **Q4: What does a channel token typically represent? Is it possible to visualize the learned channel tokens?**
>
> **A4:** We have add visualization of codebook tokens in Appendix A1. As shown in Figure 4, our channel-
> wise tokenizers can capture image structure and local details simultaneously. Please refer to A.1 for more details.

---

> > ### Comment · Reviewer_xTUG · 2024-11-25
> >
> > Thanks for your response. The visualization of the codes looks interesting. However, I still have several concerns.
> >
> > **Abount Local Details Encoding**
> >
> > As seen from the visualization, while some of the codes concentrate on local area, it's questionable if they encode details such as textures and edges, as the resolution of these codes is small. Besides, I don't think the higher IS score proves that more details are captured by the tokenizer, since the details can be added by the decoder via adversarial training after tokenization.
> >
> > **About Updated Results**
> >
> > The authors updated the results of image generation in the revised paper, with explanation that "MaskGiT sampling is not compatible with our method" in the "Overall Rebuttal". I wonder why it's not compatible?
> >
> > **About Entropy Regularization**
> >
> > In line 173 of the revised paper, the authors changed the expression "We also adopt entropy regularization" to "We find entropy regularization is bad for our codebook learning, and do not use it." However, the results in Table 5 and Table 6 remain the same. So are these results obtained using entropy regularization or not?
> >
> > &nbsp;
> >
> > To sum up, I generally like the idea of channel-wise quantization. However, some claims in the paper lack direct and sufficient verification. Therefore, I tend to keep my rating at 5.

---

> > > ### Author Response · Authors · 2024-11-25
> > >
> > > **Q1: Abount Local Details Encoding**
> > >
> > > **A1:** First, we want to stress the uniqueness of our tokenizers. As shown in the visualization, our tokenizers can capture global and local information, simultaneously. **This is a distinct characteristics for our tokenizers.**
> > >
> > > Second, although the decoder can add image details, which image details that should be added also is controlled by the image tokens generated by our tokenizers. From this point, IS score still can prove that more details are captured by the tokenizer.
> > >
> > > **Q2: About Updated Results**
> > >
> > > **A2:** We must first point out that **the new guidance schedule is widely used by MagViT-v2, TiTok[2] and MaskBit[3]. Thus, it is fair to use the updated performance results for comparison.** If not using it, it is unfair for our method to compare with other methods.
> > >
> > > MaskGiT sampling uses a constant guidance scale during the entire sampling process. It is not compatible with good image generation.  Using a constant guidance scale encounters a trade-off issue between diversity and fidelity. Thus, the guidance schedule in [1] is widely used in many methods, which makes the model samples with more diversity at early steps while samples with higher fidelity at late steps. **Based on these, it is more reasonable to claim MaskGiT sampling is not compatible with masked-based image generation, not only our method.**
> > >
> > > **Q3: About Entropy Regularization**
> > >
> > > **A3:** We are sorry to confuse you. Only the experiment using codebook size 2^17=131072 is without entropy regularization. We will revise the paper.

---

> > > > ### Comment · Reviewer_xTUG · 2024-11-25
> > > >
> > > > The answer for Q1 does not convince me. As for Q3, the author presents a totally opposite statement before and after revision. According to the authors' response, this statement still needs to be revised. Therefore, I am afraid that the paper is not well prepared enough.

---

> > > > > ### Author Response · Authors · 2024-11-25
> > > > >
> > > > > response for Q1: Thanks for your comments. We will improve it in the future.
> > > > >
> > > > > response for Q3: Only the experiment using codebook size 2^17=131072 is without entropy regularization. This demonstrates that that channel-wise quantization is the main factor behind high codebook usage. It is completely validated. Sure, in your opinion, this paper needs to be revised. However, **there is no objection to the claim: channel-wise quantization is the main factor behind high codebook usage.**

---

### Official Review · Reviewer_FnTD · 2024-11-03

**Soundness:** 1
**Presentation:** 3
**Contribution:** 1
**Rating:** 3
**Confidence:** 4

**Summary:**

This paper presents an alternative to standard VQ-VAE, which typically quantizes each spatial position as a token. Instead, this work quantizes image features along the channel dimension into discrete codes.

For comparison, in a standard VQ-VAE with a final feature dimension of C*H*W, there would be H*W tokens, whereas this work produces C tokens.

Advantages:
1. The paper is clearly written and easy to understand
2. The metrics appear reasonable

Disadvantages:
1. Channel-wise quantization lacks theoretical justification and Intuitive rationality.
   Without spatial-based quantization, the model training appears to lose its causal nature
2. As the model scales up, the number of tokens would need to increase, potentially making learning more difficult
3. The resulting tokenizer becomes incompatible when image resolution changes

I believe the motivation behind this idea is fundamentally flawed, which leads to several issues:
- Limited generalizability
- Sequence length problems
- Modeling methodology concerns

Therefore, I lean towards a negative assessment of this work.

**Strengths:**

Advantages:
1. The paper is clearly written and easy to understand
2. The metrics appear reasonable

**Weaknesses:**

Disadvantages:
1. Channel-wise quantization lacks theoretical justification and Intuitive rationality.
   Without spatial-based quantization, the model training appears to lose its causal nature
2. As the model scales up, the number of tokens would need to increase, potentially making learning more difficult
3. The resulting tokenizer becomes incompatible when image resolution changes

I believe the motivation behind this idea is fundamentally flawed, which leads to several issues:
- Limited generalizability
- Sequence length problems
- Modeling methodology concerns

Therefore, I lean towards a negative assessment of this work.

**Questions:**

see above

---

> ### Author Response · Authors · 2024-11-14
> **rebuttal**
>
> Thank you for your careful review and helpful suggestions!
>
> **Q1: Channel-wise quantization lacks theoretical justification and Intuitive rationality. Without spatial-based quantization, the model training appears to lose its causal nature**
>
> **A1:**
>
> **Theoretical justification and Intuitive rationality:** As shown in section 3.2 Discussion, the spatial tokenizers pay more attention to local image areas and are easily to collapse into a limited number of visual units. If image tokens concentrate on larger image areas, image tokens can be more diverse, resulting in high usage for codebook. For channel-wise tokenizer, it forces image tokens to focus on large image areas, due to its large receptive filed. Thus, channel-wise tokenizer is an alternative to achieve high usage of codebook.
>
> **causal nature:** I disagree with this statement. As demonstrated in MAR [1], masked generative models used by  our method are generalized autoregressive (AR) models, having causal nature. As shown in experiments, our method shows high performance on image generation. It demonstrates our image tokens are compatible with these generalized AR models. From this point, our channel-wise tokenizer do not lose causal nature.  For strict AR modeling, we leave it for future work.
>
> **Q2: As the model scales up, the number of tokens would need to increase, potentially making learning more difficult.**
>
> **A2:** We evaluate the scalability of our method from two perspectives. First, based on 16384 codebook size, we train a large model (1.0B) to validate the scalability of our method. Our model achieves 1.91 FID and 344.9 IS, outperforming VAR (2.0B) on these two metrics. Second, we conduct a experiment using codebook size 2^17=131072 without entropy regularization. The tokenizer obtains 1.33 rFID and 100\% usage on ImageNet. This result validates the scalability of our channel-wise tokenizer on codebook size.
>
> **Q3: The resulting tokenizer becomes incompatible when image resolution changes**
>
> **A3:** **Although a model is trained for one specific image resolution, we can train different models for any image resolution.**  Our method is not limited to one specific image resolution, as shown in Table 2 and Table 3.
>
> [1] Li, Tianhong, et al. "Autoregressive Image Generation without Vector Quantization." arXiv preprint arXiv:2406.11838 (2024).

---

> > ### Comment · Reviewer_FnTD · 2024-11-25
> > **Not convinced by the authors' rebuttal**
> >
> > In the limitation section, the authors state "First, our channel-wise tokenizer needs to be trained separately for different image resolution. Conversely, spatial tokenizers are trained in a low resolution
> > once but directly used for various high image resolution, though the transfer performance is not". But in the rebuttal, you mention "Our method is not limited to one specific image resolution", we think it is inconsistent. I am not convinced by the authors' rebuttal.

---

> > > ### Author Response · Authors · 2024-11-25
> > >
> > > "Our channel-wise tokenizer needs to be trained separately for different image resolution." means we should train different models for different image resolution. This is not inconsistent with "Our method is not limited to one specific image resolution." Our method means the channel-wise quantization, not the tokenizer trained on a specific image resolution.

---

> > > > ### Author Response · Authors · 2024-11-25
> > > >
> > > > We admit that the resulting tokenizer is specific to one image resolution. Thus, our method needs to train different models for different images. It is a limit of our method. But we stress the contribution of our method: 1) it is a alternative to scaling codebook; 2) it can capture global and local information simultaneously. 3) the performance on image generation is superior compared with VAR. We hope the reviewers re-evaluate the contribution of our method.

---

### Official Review · Reviewer_Xmae · 2024-11-04

**Soundness:** 3
**Presentation:** 3
**Contribution:** 2
**Rating:** 5
**Confidence:** 4

**Summary:**

This paper provides an interesting perspective on image compression through learning a vector quantized autoencoder. Typically, we tokenize images spatially. That is, every (i,j) for i over h and j over w within an hxwxc latent space, each position is mapped to a token in a learned codebook. In contrast, this paper proposes to tokenize along the channel dimension of the latent space within the AE. This results in tokens that capture the global image along a latent channel dimension. Results are conducted on class conditional and text conditional image synthesis with various ablations and analysis.

**Strengths:**

- The paper tackles an interesting approach to tokenize along the channel dimension within a VQ AE for images.
- The paper is well-structured and written.
- The approach is easy to understand and to follow.
- Comparison against many relevant models (albeit not exactly fairly if I understood correctly; see below for details).

**Weaknesses:**

My main concerns are regarding fair evaluation wrt compression ratios, more analysis on various token dimensions and exploration of usage of channel wise tokenization in downstream diffusion and AR tasks.

- Since each token in channel space captures complete global information, configurations such as [4,64,64] result in only four tokens of size 64x64 each, rather than 64x64 tokens of size 4. This impacts discussions on code embedding size and codebook use, which are misleading because they affect the compression ratio. I suggest comparing against a fixed compression budget with various compression ratios rather than focusing solely on code embedding sizes (which does not give the full picture).
- Adding to the above, the token count in Table 5 is potentially misleading, as the models use different compression ratios. If I understand correctly, the comparison is between VQAN using 4x16x16 tokens (4x256) and the proposed model with 256x16x16 tokens (256x256).
- The ablation study for token dimensions is incomplete. Experiments within the range [8, 256] are needed to understand scaling behavior better.
- Tokenizing *global information* along the channel dimension implies a significant correlation among tokens, meaning each image is learned as a whole rather than in parts. Consequently, *C* tokens per image are kind of memorized, making it difficult to repurpose tokens for other images due to global encoding. This also explains the high overall usage of the codebook, as tokens are not easily reusable.
- Adding codebook size, embedding dimensions, and compression ratios to the tables would improve comparability.
- In Table 5, why is rFID significantly better while PSNR is worse, even though SSIM is better? I would expect a consistent trade-off between perceptual- and pixel-wise metrics.
- Exploring channel-wise quantization in autoregressive (AR) or diffusion tasks could be insightful. I assume it may not perform as well for AR tasks, as the AR function would need to predict the entire global image in one step rather than progressively building it up from parts.
- The claims regarding low similarity between channel tokens, efficient modeling of image structure, and strong representational capacity lack clear definition and verification.
- I have ignored going into details regarding quantitative results mainly because the issue of fair evaluation is not cleared yet. As of now, sometimes the model is better, sometimes worse, and it is unclear why and when one would want to choose this method over the spatial tokenization.

**Questions:**

NA

---

> ### Author Response · Authors · 2024-11-14
> **rebuttal**
>
> Thank you for your careful review and helpful suggestions!
>
> **Q1&Q2: Fair evaluation wrt compression ratios, more analysis on various token dimensions. The token count in Table 5 is potentially misleading, as the models use different compression ratios.**
>
> **A1&A2:** In table 5, for VAGAN and MaskGiT, they use 256x16x16 tokens (256x256). The usage of tokens is the same as Ours. The comparison is fair. For LlamaGen, it use 8x16x16 tokens. We use this type tokens for LlamaGen, due to its higher performance. If LlamaGen uses 256x16x16 tokens, the rFID is 9.21, which is worse than ours (1.64). We have updated the Table 5 to clarify this issue.
> For Table 2, Table 3 and Table 4, the compression ratios are 16, 32, 16, respectively. Thus, tokens used by our model are all 256 (16x16). The comparison is fair.
>
> **Q3: The ablation study for token dimensions is incomplete. Experiments within the range [8, 256] are needed to understand scaling behavior better.**
>
> **A3:** We add the result of token dim 32 into Table 6 (a). In this setting, LlamaGen achieves 3.22 rFID and 20.9\% usage. This result is worse than one with dim 8. Combining with other results, LlamaGen can not perform better than one with dim 8 in range (8, 256].
>
> **Q4: Tokenizing global information along the channel dimension implies a significant correlation among tokens, meaning each image is learned as a whole rather than in parts. Consequently, C tokens per image are kind of memorized, making it difficult to repurpose tokens for other images due to global encoding. This also explains the high overall usage of the codebook, as tokens are not easily reusable.**
>
> **A4:** First, for our method, due to large receptive fields, image tokens are forced to capture large image areas and more diverse than ones generated by spatial tokenizers. This is the methodology of our channel-wise tokenizer. **Second, I disagree with the difficult reusable for our image tokens.** As shown in Table 6 (b), our tokenizer with 1024 codebook size can achieve 2.25 rFID. This means we can use 1024 tokens to represent 50000 images (ImageNet eval set) at least. Since an image uses 256 tokens, 50000 images means 12.8M image tokens. **If our 1024 tokens are not reusable, how to represent 12.8M image tokens?**
>
> **Q5: Adding codebook size, embedding dimensions, and compression ratios to the tables would improve comparability.**
>
> **A5:** Thanks for your suggestion. We have added some necessary information into Table 5 to clarify the fair comparison.
>
> **Q6: In Table 5, why is rFID significantly better while PSNR is worse, even though SSIM is better? I would expect a consistent trade-off between perceptual- and pixel-wise metrics.**
>
> **A6:** The reason is two folds. First, the codebook size is not large enough to represent all the details of images. When we use larger codebook size(65536), PSNR is increased to 18.72. Second, the tokens representing an image is also not large enough. As shown in Table 5, with 512 tokens, PSNR is increased to 21.47. Meanwhile, rFID and SSIM are also improved largely. The improvement on all metrics shows that there is not a consistent trade-off between perceptual- and pixel-wise metrics.
>
> **Q7: Exploring channel-wise quantization in autoregressive (AR) or diffusion tasks could be insightful. I assume it may not perform as well for AR tasks, as the AR function would need to predict the entire global image in one step rather than progressively building it up from parts.**
>
> **A7:** I disagree with the assumption. For our channel-wise tokenizer, it forces image tokens to capture large image areas but not the whole image. If image tokens generated by our tokenizer represent the whole image, they are not reusable in other images. As shown in A4, tokens in codebook are reusable, it challenges the assumption.
>
> **Q8: The claims regarding low similarity between channel tokens, efficient modeling of image structure, and strong representational capacity lack clear definition and verification.**
>
> **A8:**
>
> **low similarity between channel tokens**
>
> This is validated by our 100\% codebook usage under different codebook size, especially 2^17 codebook size. If there is high similarity between channel tokens, codebook usage can not reach 100\%.
>
> **efficient modeling of image structure**
>
> This is validated by high SSIM score, demonstrated in Table 5. SSIM is sensitive to image structure changes. High SSIM means efficient modeling of image structure. We have discussed it in “Comparisons with other image tokenizers”.
>
> **strong representational capacity**
>
> First, for image reconstruction, our tokenizer outperforms baseline (LlamaGen) on rFID and codebook usage, as shown in Figure 1, Table 5 and Table 6 (a). Second, for image generation, generators with our tokenizer outperform other generators, like VAR. These both demonstrate strong representational capacity of our tokenizer.

---

> ### Author Response · Authors · 2024-11-14
> **rebuttal**
>
> **Q9: I have ignored going into details regarding quantitative results mainly because the issue of fair evaluation is not cleared yet. As of now, sometimes the model is better, sometimes worse, and it is unclear why and when one would want to choose this method over the spatial tokenization.**
>
> **A9:** We clarify the issue of fair evaluation in A1&A2. With the updated results, our method based on channel-wise tokenizer outperforms other generators, like VAR. Besides, our tokenizer can reach 100\% codebook usage, especially 2^17 codebook size. Based on high generation performance and 100\% codebook usage, our tokenizers is an alternative to spatial tokenization.

---

### Author Response · Authors · 2024-11-14
**Overall Rebuttal**

We thanks all the reviewers for their hard work! We are very honored that our work has been recognized for: **1) interesting approch(Reviewer Xmae, xTUG); 2) novel idea (Reviewer xTUG); 3) clearly written (Reviewer Xmae, FnTD, xTUG, 6xWH).**

## Updated Performance Results
We find MaskGiT sampling used by our method is out-of-date and is not compatible with our method. Thus, we adopt the guidance schedule from MDTv2 [1]. This guidance schedule is widely used by MagViT-v2, TiTok[2] and MaskBit[3]. Thus, it is fair to use the updated performance results for comparison.  We have updated performance results in the paper. Here, we list some critical results to show the effectiveness of our method. **Our method outperforms VAR on FID and IS by a large margin, especially on ImageNet 512x512**

**ImagNet 256x256**

+: codebook size is 65536.

|   Models    | FID | IS | #Para| #step|
| ----------- | ----------- |----------- |----------- |----------- |
|VAR-d30 | 1.97 | 334.7 | 2.0B | 10 |
|MagViT-2 | **1.78** | 319.4 | 307M | 64 |
|Ours-B | 2.77 | 305.3 | 305M | 10 |
|Ours-L | 2.46 | 302.5 | 634M | 10 |
|Ours-H |2.39 |338.2 | 1.0B | 10|
|Ours-B* | 2.21 | 301.2 | 305M | 64 |
|Ours-L* | 2.02 | 323.4 | 634M | 64|
|Ours-H* |1.91 | **344.9**| 1.0B | 64 |
|Ours-L+ | 1.87 | 320.4 | 634M | 64 |

**ImagNet 512x512**
|   Models    | FID | IS | #Para| #step|
| ----------- | ----------- |----------- |----------- |----------- |
|VAR-d36-s | 2.63 | 303.2 | >2.0B | 10 |
|MagViT-2 | **1.91** | 324.3 | 307M | 64 |
|Ours-B | 2.68 | 318.5 | 305M | 10 |
|Ours-L  | 2.46 | 336.4 | 634M | 10 |
|Ours-B* | 2.22 | 323.4 | 305M | 64 |
|Ours-L* | 2.01 | **341.5**| 634M | 64|

**MS-COCO 256x256**
|   Models    | FID | #step|
| ----------- | ----------- |----------- |
|U-ViT-S/2 | 5.95 | - |
|Ours-L | 6.40 | 10 |
|Ours-L*  | **5.85** | 64 |

[1] Gao, Shanghua, et al. "Masked diffusion transformer is a strong image synthesizer." Proceedings of the IEEE/CVF International Conference on Computer Vision. 2023.

[2] Yu, Qihang, et al. "An Image is Worth 32 Tokens for Reconstruction and Generation." arXiv preprint arXiv:2406.07550 (2024).

[3] Weber, Mark, et al. "Maskbit: Embedding-free image generation via bit tokens." arXiv preprint arXiv:2409.16211 (2024).

## Methodology
As shown in section 3.2 Discussion, the spatial tokenizers pay more attention to local image areas and are easily to collapse into a limited number of visual units. If image tokens concentrate on larger image areas, image tokens can be more diverse, resulting in high usage for codebook. For our method, due to large receptive fields, image tokens are forced to capture large image areas and more diverse than ones generated by spatial tokenizers. Meanwhile, for reconstruct images in high quality, our tokenizers must learn local details. We have add visualization of codebook tokens in Appendix A1. As shown in Figure 4, our channel-wise tokenizers can capture image structure and local details simultaneously. Please refer to A.1 for more details.

---

### Author Response · Authors · 2024-11-25
**Looking forward to your feedback**

Dear reviewers,

Thank you for your valuable comments regarding our submission.

We have posted our responses and hope to address your concerns about the motivation of our method. We also show the superior results of out method with widely used sampling scheme, demonstrating the effectiveness of our proposed method.

Could you please take a look at our responses and possibly re-evaluate our work based on the additional input? Thank you!

Best regards,

Authors

---

### Meta-Review · Area_Chair_1XPU · 2024-12-16

**Metareview:**

This paper introduces channel-wise quantization, which discretizes image features along channels as an alternative to standard spatial tokenizers. The paper received negative feedback from all reviewers. While the authors’ rebuttal adequately addressed some concerns, several major issues remain unresolved: (1) the performance improvement is somewhat marginal and insufficient to outweigh the limitation of fixed image resolution, (2) the argument regarding the global-local information captured by the proposed tokenizer is not convincingly presented, and (3) the experiments are limited in demonstrating the claimed improvement in codebook usage over spatial tokenizers. Overall, the AC believes the paper presents an interesting alternative to spatial tokenizers, with some promising results (e.g., Figure 4). However, the AC agrees with the reviewers that the paper requires further improvements in experiments and presentation, and is not ready for publication at this time. Therefore, the AC recommends rejection.

**Additional Comments On Reviewer Discussion:**

The common major concerns raised by the reviewers (Xmae, FnTD, 6xWH) focused on the practicality of the method. Specifically, the proposed tokenizer restricts the learned codebook from generalizing across various spatial resolutions, while the performance improvements are not significant. Additionally, the reviewers noted that some of the authors' arguments—such as those regarding the global-local information captured by the learned codebooks and the improvement in codebook utilization—require stronger support through more carefully controlled experiments.

---

### Decision · Program_Chairs · 2025-01-22

Reject